# The Emerging Functions of Circular RNAs in Bladder Cancer

**DOI:** 10.3390/cancers13184618

**Published:** 2021-09-15

**Authors:** Kai Sun, Di Wang, Burton B. Yang, Jian Ma

**Affiliations:** 1Urology Department, The Affiliated Yantai Yuhuangding Hospital of Qingdao University, Yantai 264000, Shandong, China; 2019026330@qdu.edu.cn (K.S.); 2020026276@qdu.edu.cn (D.W.); 2Department of Laboratory Medicine and Pathobiology, Sunnybrook Research Institute, University of Toronto, Research Building, 2075 Bayview Avenue, Toronto, ON M4N 3M5, Canada

**Keywords:** circular RNA, bladder cancer, biomarker, signaling pathways, drug resistance

## Abstract

**Simple Summary:**

The role of circular RNAs has made breakthroughs in understanding the mechanisms of tumor development. Bladder cancer has an increasing incidence, high recurrence rate, high metastatic potential, poor prognosis, and susceptibility to chemotherapy resistance. Thus, it is essential to identify molecules related to the tumorigenesis of bladder cancer. In this review, we summarize current knowledge about the expression of circular RNAs in bladder cancer and their implications in vesical carcinogenesis. We further discuss the limitations of existing studies and provide an outlook for future studies in the hopes of better revealing the association between circular RNAs and bladder cancer.

**Abstract:**

Bladder cancer (BC) is among the top ten most common cancer types worldwide and is a serious threat to human health. Circular RNAs (circRNAs) are a new class of non-coding RNAs generated by covalently closed loops through back-splicing. As an emerging research hotspot, circRNAs have attracted considerable attention due to their high conservation, stability, abundance, and specificity of tissue development. Accumulating evidence has revealed different form of circRNAs are closely related to the malignant phenotype, prognosis and chemotherapy resistance of BC, suggesting that different circRNAs may be promising biomarkers and have therapeutic significance in BC. The intention of this review is to summarize the mechanisms of circRNA-mediated BC progression and their diagnostic and prognostic value as biomarkers, as well as to further explore their roles in chemotherapy resistance.

## 1. Introduction

Bladder cancer (BC) is reported to be the 10th most prevalent human malignancy worldwide, with an estimated 400,000 new cases diagnosed and more than 165,000 deaths annually [1,2,3].Currently, new technologies are being developed to improve the detection of BC, and new treatment options are being offered in the guidelines [4,5].The first-line treatment for BC is surgical section. In addition to traditional surgery, as well as chemotherapy and radiation, immunotherapy has been applied to treat BC patients [6,7]. However, including after radical cystectomy, distant metastases still exist in certain patients [4]. For advanced or metastatic BC patients, chemotherapy with cisplatin is the first-line treatment, and the efficacy is limited by chemotherapy resistance [8,9]. Tumor recurrence, metastasis and resistance to chemotherapy drugs make the overall therapeutic effects unsatisfactory, with a low five-year survival rate [10,11]. Appropriate molecular biomarkers can provide accurate information for BC staging to improve its early diagnosis and treatment efficacy. The molecular biological mechanisms underlying the tumorigenesis, progression, and chemoresistance of BC have attracted extensive attention from researchers, and new BC-related biomarkers are critical to improve the diagnosis and prognosis of BC patients [12,13,14]. 

BC tumorigenesis is a complicated process involving genetic mutations and dysregulation of epigenetic pathways. Epigenetic changes in BC, such as non-coding RNAs (NcRNAs) [15], DNA methylation [16], and histone modifications [17], have been extensively studied. Numerous ncRNAs have been demonstrated to participate in tumor initiation and progression. Therefore, an emerging field of clinical research is to explore the feasibility of targeting ncRNAs.

Circular RNAs (circRNAs) are single-stranded, closed-loop structures lacking 5′ caps and 3′ tails of linear RNA, which enable them to resist the degradation of Ribonuclease R (RNase R) and thus are more stable than linear RNA [18,19,20]. CircRNAs are an important part of non-coding RNA that are emerging as key new members of the gene regulatory milieu. The regulatory functions of circRNAs in physiological and pathological environments have been the focus of previous studies [21,22]. Recently, the role of circRNAs has been revealed in a variety of cancers, such as prostate cancer, glioma, breast cancer, colorectal cancer, and more [23,24,25,26,27,28,29]. A large number of studies have shown that circRNAs play a significant role in the progression of BC, including cell proliferation, migration and invasion, metastasis, cell cycle, apoptosis and drug resistance [30,31,32]. Furthermore, it has also been demonstrated that the abnormal expression of circRNA is related to its pathological characteristics in bladder cancer tissue, which can be used as a potential biomarker for early screening, diagnosis and prognosis of bladder cancer. CircRNAs possess potential modes of specific action, serving as sponges for miRNAs and RNA-binding proteins, or acting as transcriptional regulators. The aim of this paper is to discuss the latest knowledge on the role of circRNAs in bladder carcinogenesis, including proliferation, invasion, metastasis, and therapeutic resistance, and to propose circRNAs that canserve as ideal biomarkers and/or therapeutic targets.

## 2. Overview of circRNAs

### 2.1. Biogenesis and Classification of circRNAs

Circular transcripts, first discovered in 1976 in a plant viroid, attracted little attention at the time and were treated as abnormal by-products or “splice noise” with low abundance and low functional potential [33,34,35,36]. Due to the development of high-throughput screening technology, more types of circRNAs have been identified in a variety of species and cell lines [22,37,38,39,40]. The expression level of circRNAs is relatively rich and highly conserved. The expression level of circRNAs is relatively common in eukaryotic cells and varies greatly due to the specificity of tissue and development stage [41,42,43].

In the light of composition and biogenesis mechanisms of RNA, circRNAs can be divided into four specific groups: exonic circRNAs (EcircRNAs), exon–intron circRNAs (EIciRNAs), circular intronic RNAs (CiRNAs), and intergenic circRNAs [41,44]. EcircRNAs containing exons are only produced in a course known as exon skip events or back-splicing circularization [45]. Lariat-driven circularization is a type of exon skipping process. Pre-mRNA splicing is commonly known to be a two-step ester exchange reaction [46]. The adenosine hydroxyl group located at the 2′ branch point within the intron that is to be spliced attacks the intron upstream of the 5′ end in a nucleophilic manner. This produces a lariat intermediate closed covalently through the 2′ to 5′ phosphodiester bond. Then, the 3′ hydroxyl group of the upstream exon is free to attack the 5′ phosphate of the downstream exon, splicing out the intron completely and allowing the exons to be attached as a linear coding sequence [46]. After being processed through lariat model, pre-mRNAs form a lariat intermediate containing exons with the help of circular introns during internal splicing [47,48]. In the lariat intermediate, the 5′ terminal donor of an exon combines with the 3′ terminal acceptor of another exon and then forms an EcircRNA by eliminating introns between exons. Back-splicing circularization is another significant process to generate EcircRNAs. During the course of back-splicing circularization, the downstream 5′ terminal of an exon unites the upstream 3′ terminal of another exon [49,50]. In addition, a circular RNA can also be generated from a single exon, where the 5′ terminal of one exon is connected to the 3′ terminal of the same exon. Intron pairing-driven circularization is a widespread back-splicing mechanism, which is a process of circularization induced by reverse complementary sequences, thus generating EIciRNA or EcircRNA by removing introns [51,52]. Another back-splicing mechanism is RBP-driven circularization, beginning with back-splicing, and then relying on RBPs, flanking introns bind tightly to each other to form a circular RNA [53,54]. However, the formation of CiRNA depends on conserved sequences near the spliceosome [55]. Specifically, lariat introns can be produced by the combination of 3′ splice receptors and 5′ splice donors, and lariat introns can avoid degradation by debranching enzymes with the aid of conserved sequence constituted by 7 nt GU-rich elements at 5′ splice site and 11 nt C-rich elements at branch point site [56] (Figure 1).

### 2.2. CircRNA Identification and Database for circRNA Research

High-throughput RNA-seq, RT-PCR/qPCR, Northern blot, circRNA microarray and other tools have been widely applied to identify and quantify circRNAs [57]. The presence and amount of circRNAs were revealed using high-throughput RNA-seq, which is equipped with next-generation sequencing in combination with ribosomal RNA depletion [58]. For known circRNAs, qRT-PCR using divergent primers and Northern blot using reverse splicing sequence- specific probes can be used to verify their presence and detect their quantities. CircRNAs can be specifically identified and quantified by circRNA microarray by binding sequence specific probes of circular junctions and external nuclease linear RNA consumption [59].

To analyze the information, regulatory networks and roles of circRNAs in diseases and other physiological processes, databases including Circbase, CIRC pedia V2 and Deepbase 2.0 containing vast quantities of circRNAs and relevant details about diverse species have been established [60,61,62]. The transcriptional regulation information of circRNAs can be supplied by the TRCirc database, and the sequencing results can be easily analyzed in the CirclncRNAnet database [63,64].The network relationships between partial miRNAs and circRNAs as well as between proteins and circRNAs have already been elucidated by Starbase v2.0, CircInteractome and other databases, which provide a great assistance to study the functions of circRNAs [65,66,67]. CircRNADb and CSCD databases for analyzing protein-coding capabilities have also been founded [68,69]. Furthermore, certain clinically relevant patient information is available from Circ2Traits and CircRNA Disease databases, providing clues to delve into the potential of circRNAs as biomarkers in certain diseases [70,71]. With the application of more techniques to identify circRNAs and the continuous improvement of databases, the roles of circRNAs will be more fully elucidated.

### 2.3. Functions of circRNAs

CircRNAs exert their activities at various levels. (a) CircRNAs have been identified as critical regulators of the major signaling pathways involved in cancer progression [72]. Dysregulation of miRNA-mediated mRNA and correlative signaling pathways are closely interrelated to cancer progression and therapeutic resistance. Numerous studies have reported that circRNAs can function as miRNA sponges or competing endogenous RNA (ceRNAs) to repress miRNA activities [73,74,75,76,77]. As a result, the expression of target genes is upregulated. CircRNA ciRS-7 was the most typical one which possesses more than 70 conserved miRNA targets [78]. CiRS-7 can increase the expression of targets of miR-7 via sponging miR-7. (b) The adjustment of circRNAs at transcriptional level may be controlled by intron sequence circRNA. Located in the nucleus, CiRNA and EIciRNAs may regulate the expression of their associated protein at the level of transcription and post-transcription [59]. Dominantly enriched in its parent genes transcriptional site, ci-ankrd52 was reported to have a positive effect on RNA pol II transcription and serve as a positive regulator for its parent gene transcription [55]. (c) CircRNAs were identified to combine with certain proteins to form particular circRNA-protein complexes (circRNPs), which can modulate the subcellular localization of proteins, the action of associated proteins and the transcription of parental genes. Ashwal-Fluss et al. reported that conservative binding sites bonded with MBL were positioned in circMBL and its flanking introns [79]. The level of circMBL biosynthesis depends on the degree of binding between MBL and its binding site. (d) Most circRNAs are derived from exons of pre-mRNA and were once thought to have no translational capability, but increasing evidence shows that circRNAs have great coding potential in a cap-independent way [80]. CircSHPRH was identified to encode a 146 amino acids protein which inhibits tumor growth by stopping ubiquitin proteasome-mediated degradation of SHPRH protein in glioma [81]. In addition to regulating basic biological processes, circRNAs also play critical roles in the progression of different types of cancers [82,83,84,85,86,87,88], cardiovascular disorders [89,90,91], neuronal degenerative diseases [92,93,94,95], and other physiological conditions [96,97,98,99,100,101]. Due to the high stability of circRNAs, we believe that they can make monumental contributions to the diagnosis and treatment of diseases.

## 3. Expression and Biological Functions of circRNAs in BC

### 3.1. Abnormal Expression of circRNAs in BC

There is growing evidence that abnormal expression of circRNAs is associated with the development of BC. Several high-throughput experiments have shown that circRNA expression profiles are dysregulated in BC. Li et al. have characterized 316 differentially expressed circRNAs in high grade BC tissues, compared to adjacent non-cancerous ones; 205 circRNAs were found to be upregulated, while 111 were downregulated [102]. Shen et al. also analyzed the differential gene expression of normal bladder tissues and paired tumor tissues and identified 5578 upregulated and 5833 downregulated circRNAs. By RNA sequencing from four pairs of bladder cancer tissues, Li et al. found transcripts of 59 differentially expressed circRNAs. Compared with adjacent tissues, 7 were upregulated and 52 were downregulated [102]. Li et al. analyzed abnormal circRNAs expression between BC and adjacent non neoplastic bladder tissues by circRNA microarray and identified 512 differentially expressed circRNAs (340 upregulated, 172 downregulated) [103]. On the basis of validation experiments, we have identified and analyzed a variety of specific circRNAs, suggesting that aberrant expression of circRNAs has potential therapeutic value.

### 3.2. circRNAs Regulate Proliferation of BC

#### 3.2.1. Oncogenic circRNAs in BC

A study from Yang et al. indicated that circUVRAG was significantly upregulated in tissues and cell lines of BC, and its knockdown dramatically inhibited cell proliferation via promoting miR-223, resulting in repression of FGFR2 [67] (Figure 2). CircRGNEF consists of 2 exons from the RGNEF gene and affected progression of BC cells via sponging miR-548, subsequently upregulating KIF2C levels [104]. Furthermore, dysregulation of the circRNA-mediated Tgf-β2/smad3 signaling pathway also participated in progression of BC. For instance, Su et al. found that circRIP2 enhanced BC progression via the Tgf-β2/smad3 signaling pathway by sponging miR-1305 [105]. Similarly, Mao et al. demonstrated that hsa_circ_0068871 was upregulated in BC and promoted BC proliferation and apoptosis through FGFR3-induced activation of STAT3 pathway by sponging miR-181a-5p [106]. Sponging miR-145-5p, the overexpressed circCEP128 was demonstrated to promote BC proliferation and inhibit apoptosis via modulating SOX11 [107]. CircDOCK1 was reported to be significantly increased in BC tissues, and its knockdown dramatically inhibited the progression of EJ-m3 and 5673 BC lines through upregulating the expression of miR-132-3p [108]. Has_circ_0068307 markedly upregulated and promoted BC cells progression through miR-147/c-Myc pathway [109]. Additionally, circ_0008532 promoted BC growth through sponging miR-155-5p and miR-330-5p, subsequently increasing MTGR1 expression [110] (Table 1).

#### 3.2.2. Anti-Oncogenic circRNAs in BC

Li et al. found an antitumor circRNA, circ-FOXO3, which is produced from the members of the fork-head family and found that circ-FOXO3 plays antitumor roles in BC by regulating the miR-9-5p/TGFBR2 axis [111]. Wang et al. revealed that circ-FOXO3 accelerated the apoptosis of BC cells through direct interaction with miR-191-5p [112]. Sponging miR-182-5p, the circBCRC-3 was proved to suppress proliferation by promoting the miR-182-5p-oriented 3′UTR activity of p27 [113]. Li et al. indicated that circCdr1as was significantly downregulated in BC specimens, and its overexpression dramatically inhibited cell proliferation via promoting miR-135a [114]. CircNR3C1was composed of end-to-end splicing of the exon-2 from the NR3C1 gene and inhibited proliferation of BC cells via sponging miR-27a-3p effectively, subsequently downregulating cyclin D1 levels [115]. CircPTPRA, which originated from the exon 8 and 9 of the PTPRA gene, was identified to sponge miR-636 to increase the expression of KLF9, suppressing proliferation of BC cells [116]. CircSLC8A1 regulated the PI3K-AKT pathway to repress BC progression via the miR-130b and miR-494/PTEN axis [117]. Yang et al. revealed that circ-ITCH inhibited the proliferative biological behaviors of BC via circ-ITCH/miR-17, miR-224/p21, PTEN axis [118]. Zeng et al. found that circRNA BCRC4 down-expressed in BC tissues and cell lines, and its forced expression inhibited viability and promoted apoptosis of UMUC3 and T24T cells through circBCRC4/microRNA-101/EZH2 signaling [119] (Table 1).

### 3.3. circRNAs Regulate Metastasis of BC

#### 3.3.1. Oncogenic circRNAs in BC Migration or Invasion

CircRNA VANGL1 may sponge various miRNAs, including miR-1184 and miR-605-3p. Yang et al. identified miR-1184 as a target of circVANGL1, while miR-1184 targeted IGFBP2, suggesting that circVANGL1 promoted bladder cancer invasion and migration through the circVANGL1/miR-1184/IGFBP2 network [120]. Another paper proved that overexpression of circVANGL1 promoted migration and invasion of BC cells by sponging miR-605-3p upregulating VANGL1 level [121]. Sun et al. stated that overexpression of circCEP128 may sponge miR-145-5p and upregulate MYD88 through MAPK signaling pathway to promote BC progression [122]. Additionally, Sun et al. identified circ_0058063, which also sponged miR-145-5p, was upregulated in BC and promoted migration but impaired cell apoptosis by regulating CDK6 expression [123]. Liang et al. suggested that circ_0058063 served as a sponge of miR-486-3p to block cell death and promote cell invasion by regulating FOXP4 expression [124]. CDK6 is also a target protein of miR-107, and circTCF25 can be used as a sponge for miR-107 and miR-103-3p to promote migration [125]. CircTFRC was also a sponge for miR-107, and the knockdown of circTFRC may decelerate invasion of BC cells by inhibiting TFRC [126]. Zhang et al. verified that circINTS4 promoted BC cell migration and cell cycle progression via promoting the NFκB signaling pathway and restraining P38 MAPK signaling pathway in a CARMA3-mediated manner [127]. Lu et al. demonstrated that circKIF4A was upregulated in BC cell lines, and its overexpression facilitated BC migration and metastatic ability through NOTCH2-induced activation of PI3K-AKT pathway by sponging miR-375 and miR-1231 [128] (Table 1).

#### 3.3.2. Anti-Oncogenic circRNAs in BC Migration or Invasion

Circ_0071662, a circinate product of TPPP transcript, was confirmed to repress invasive biological behaviors by sponging miR-146b-3p, and then boosting HPGD and NF2 expression [129]. Hsa_circ_0001546 (circFAM114A2), which derived from the FAM114A2 gene and was spliced by exons 2–4, was shown to possess potential biological roles in inhibiting migration and invasion of BC via a circFAM114A2/miR-762/ΔNP63 axis [130]. Zhang et al. reported that hsa_circ_0091017 was remarkably downregulated in BC cell lines and tissues, and the inhibitory effect on the malignant phenotype may be reversed by overexpression of microRNA-589-5p [131]. By sponging miR-197-3p, circular RNA hsa_circ_0002024 was found to suppress migratory and invasive biological behaviors of BC cells [132]. Liu et al. elucidated that circUBXN7 act as a ceRNA of miR-1247-3p to enhance B4GALT3 expression, thus repressing cell invasion and viability [133]. Another study showed that circFNDC3B inhibited cancer cell migration and invasion by binding miR-1178-3p, which targeted the oncogene G3BP2, thereby suppressing the downstream SRC/FAK pathway [145]. EGFR pathway was connected to malignant biological behaviors. A study from Wu et al. revealed that circ_0023642 promoted invasion through acting as a miR-490-5p sponge via the EGFR pathway [135]. In addition, they also found that estrogen receptor alpha (ERα) altered circ_0023642 levels by regulating the expression of its host gene, UVRAG, uncovering upstream regulatory mechanism. Xie et al. described circPTPRA as a novel tumor suppressor which repressed cancer invasion and migration via endogenous inhibition of the recognition of IGF2BP1 from m6A-modified RNAs [136] (Table 1) (Figure 2).

#### 3.3.3. CircRNAs in Regulation of EMT

Multiple researchers reported that circRNAs affected BC metastasis by modulating epithelial-mesenchymal transition (EMT) (Figure 2), which was involved in malignant biological functions of tumors [146]. One study showed that circ_0006332 sponged miR-143 to increase the expression of its target MYBL and consequently promoted EMT in bladder tumors [137]. CircRIMS1 was highly expressed in BC, and its knockdown significantly suppressed expression of N-cadherin and vimentin, and enhanced expression of E-cadherin, suggesting suppression of EMT process [32]. Meanwhile, circPRMT5 was dramatically upregulated in BC and was positively correlated with pathological stage, and it significantly increased the aggressiveness ability of BC cells by specifically sponging miR-30c and promoting EMT [138]. Zhong et al. indicated that circ-MYLK promoted BC malignancy via facilitating VEGFA/VEGFR2 signaling and its downstream Ras/ERK pathway by sponging miR-29a, which dramatically boosted EMT in BC [139]. Tong et al. suggested that circ_100984/miR-432-3p axis modulated c-Jun/YBX-1/β-catenin feedback loop to influence EMT and thus promote tumor progression [140]. CircRBPMS reduced the inhibition of RAI2 by sponging miR-330-3p, thereby suppressing the ERK signaling pathway and inhibiting the EMT process to repress BC progression [141]. Tan et al. proved that circST6GALNAC6 inhibited EMT and BC metastasis partly via the miR-200a-3p/STMN1axis [142]. Overexpression of circ_0000629 resulted in a significant increase in E-cadherin expression and a remarkable decrease in Vimentin, Snail, and N-cadherin expression, which inhibited the aggressiveness of BC [143]. CircPICALM served as a sponge for miR-1265 to modulate STEAP4 and further affected the condition of pFAK-Y397 and EMT, thus inhibiting BC progression [144]. EMT was one of the most significant molecular pathways that promoted metastatic ability of cancer cells and EMT was thought as a probable target of miRNAs in cancer cells [147,148]. Considering the functional relationship between circRNAs and miRNAs, the potential roles of circRNAs in regulating EMT is worthy of further exploration (Table 1).

### 3.4. CircRNAs in BC Drug-Resistance and Chemo-Sensitization

Surgical treatment of BC remains the preferred treatment option, but chemotherapy with gemcitabine and cisplatin is the standard first-line treatment for patients with advanced or metastatic BC [149]. However, chemotherapy resistance often leads to tumor recurrence and progression. Several studies have reported the relationship between circRNAs and chemotherapeutic resistance, suggesting that circRNAs may be potential therapeutic targets [150,151,152,153,154]. A study by Yuan et al. showed that upregulation of circular RNA Cdr1as is related to cisplatin sensitivity in BC patients by upregulating expression of APAF1 via miR-1270 repression [155]. CircHIPK3, which can bind to miR-558 directly, is significantly downregulated in gemcitabine resistant cells [156]. Su et al. identified that circELP3 promotes proliferation and reduces apoptosis by adapting to hypoxic tumor microenvironment and facilitates resistance of cisplatin by targeting cancer stem-like cells [157]. Meanwhile, the upregulation of circELP3 is related to higher lymphatic metastasis and pathological stage, which implies that circELP3 can be a potential therapeutic and prognostic target for BC patients. In another study, circFNTA was found to be extremely upregulated in cisplatin resistant BC cells [158]. CircFNTA can regulate cisplatin resistance by binding to miR-370-3p and altering KRAS activity. In addition, Chen et al. also found that androgen receptors (AR) affect circFNTA levels by inhibiting RNA editing gene ADAR2, thus increasing BC invasive ability and cisplatin resistance. Furthermore, the downregulation of circ102336 may increase cisplatin sensitivity in cisplatin-resistant BC cells by altering miR-515-5p [159]. circ_000285 levels in cisplatin sensitive patients was nearly 3 times higher than that in cisplatin resistant patients, and circRNA_000285 levels in parental cells was higher than that in cisplatin resistant RT4 cells, indicating that circRNA_000285 may serve as a biomarker for BC diagnosis and chemotherapy [160]. Additionally, Huang et al. figured out that circRNA_103809 can enhance chemo-resistance of BC cells by modulating miR-516a-5p/FBXL18 axis [161] (Table 2).

A growing number of circRNAs have been shown to be implicated in chemotherapy resistance in BC. A meaningful method to achieve the reversal of drug resistance is to target these aberrantly expressed circRNAs. Drug resistance in BC can be reversed by exogenous expression of anti-oncogenic circRNAs or knockdown of oncogenic circRNAs by short hairpin RNAs (shRNAs) or small interfering RNAs (siRNAs). Drug-resistant advanced BC patients may benefit from circRNAs-based therapeutic interventions in combination with conventional chemotherapy or targeted therapy.

## 4. CircRNAs Are Potential Diagnostic and Prognostic Biomarkers of Bladder Cancer

Furthermore, circRNAs show interest as potential diagnostic and prognostic biomarkers for BC. Functional studies of circRNAs have shown that circRNAs have important clinical application value in regulating downstream target genes by acting as oncogenes or tumor suppressors. As mentioned above, circRNAs are characterized by their developmental stability, evolutionary conservation, specificity of tissue development, and abundance of tissue content, and they are widely found in blood, saliva, and urine. Therefore, circRNAs have become valuable biomarkers for the diagnosis, prognosis, and efficacy evaluation of bladder cancer.

Up until now, the relation between expression of circRNAs in tissues and clinical parameters have been widely reported. CircVANGL1, circ_0067934, circ-ASXL1, circGprc5a and circ_0001361 related to poor prognosis and clinical severity in patients with BC [121,162,163,164,165]. Meanwhile, circ_0000285, circMTO1, circLPAR1 were connected with good overall survival and good prognosis [160,166,167]. Dong et al. described the reduction of circACVR2A was related to advanced WHO grade and larger tumor size, implying circACVR2A may be a prognostic marker [168]. Moreover, upregulated circACVR2A inhibited invasion and migration through miR-626/EYA4 axis. Zhang and colleagues introduced circZFR as a cancer-independent prognostic biomarker for patients with BC [169]. Based on progression-free survival (PFS) and overall survival (OS) curves, downregulation of cirZFR significantly linked with better OS and PFS, which is opposed to circZFR upregulation in BC. Furthermore, circZFR downregulation is related to lower incidence of lymphatic metastasis. Thus, in combination with clinicopathological features, circZFR expression provides a better prognostic and diagnosis value in BC screening. A recent research using the FISH technique demonstrated the dramatic downregulation of circFUT8 in BC patients [170]. Moreover, clinicopathological results revealed that the upregulation of circFUT8 is negatively linked to the lymphatic metastasis in BC patients. Besides, the Kaplan–Meier analysis indicated circFUT8 may serve as a prognostic marker for good OS for BC patients. CircZKSCAN1 is another circRNA whose expression is evidently lower in patients at early-stage BC compared with advanced stage patients [171]. Additionally, survival analysis revealed BC patients with lower circZKSCAN1 expression suggested longer OS rates than patients with higher circZKSCAN1 expression. Besides, the upregulation of circZKSCAN1 along with the lymphatic metastasis are considered as independent prognostic factors of BC patients. As a result, circZKSCAN1 may be regarded as a biomarker for survival prediction. Yan et al. characterized circPICALM as a potential prognostic predictor for BC patients [144]. Their results showed that circPICALM is downregulated in patients at late-stage, and its reduction is linked to lower OS rates and higher TNM grades. These findings illustrate that circPICALM may be a potent prognostic indicator. Circ_0018069 is another new circRNA downregulated in BC tissues in comparison with the paired adjacent normal tissues [172]. Its deficiency in BC is notably associated with prognostic clinicopathological indices, for example, distant or lymphatic metastasis and poor OS. Based on these associations, circ_0018069 may be a valuable biomarker for prognosis and diagnosis prediction (Table 3).

The above examples idealize a blueprint for the application of circRNAs as biomarkers in the diagnosis and prognosis of BC. However, there are still issues that need to be addressed for the clinical application of circRNAs. Due to different expression in tissues, circRNAs are hard to detect in serum and plasma. In addition, the normal values and fluctuating ranges of circRNAs have not been determined, as well as whether their expressions are time-dependent. Despite these difficulties, circRNAs remain potential biomarker candidates that may offer additional diagnostic and prognostic possibilities for BC.

## 5. Relationships between circRNAs Quantities and Clinicopathologic Features in BC

CircRNAs are closely associated with many clinicopathological features in BC, including tumor size, grade, tumor number, lymph node metastasis, distant metastasis, stage, and recurrence. CircRIP2 is evidently upregulated in BC tissues, and its expression quantities are closely connected with tumor number, tumor stage, and distant metastasis [105]. Wang et al. observed that hsa_circ_100146 is significantly upregulated in BC tissues and that hsa_circ_100146 levels are associated with tumor size, tumor grade, lymph node metastasis, and tumor number [173]. Similarly, circ_0006332 is dramatically upregulated in BC tissues, and its expression quantities are positively associated with tumor grade, tumor stage, and tumor size [137]. Li et al. demonstrated that circHIPK3 levels are evidently reduced in BC tissues and that this downregulation is correlated with tumor size, tumor stage, and lymph node metastasis [174]. CircFNDC3B has been reported to have low-expression in BC tissues, and its levels are associated with invasion, tumor grade and stage, and recurrence [145]. Similarly, circPICALM has a low expression in BC tissues, and its levels in BC tissues are related with tumor size, stage, grade and lymph node metastasis [144]. In contrast, hsa_circ_102336 is upregulated in BC cell lines and tissues, and its expression quantities are related with TNM stage, tumor size, grade, and distant metastasis [159]. Circ_0137439 has been identified to be upregulated in urine samples of patients with BC and the hsa_circ_0137439 levels are associated with tumor stage, grade, and lymph node metastasis [175]. Lu et al. revealed that low circSLC8A1 expression in BC tissues is correlated with tumor size, grade, and lymph node metastasis [117]. According to Liang and colleagues, hsa_circ_0058063 is significantly upregulated in BC tissues compared with normal tissues, and its expression quantities are associated with tumor number, grade, and lymph node metastasis [124]. The expression quantities of circ-ZKSCAN1 have been revealed to be decreased in BC tissues, and low levels of circ-ZKSCAN1 are positively related with tumor size, grade, recurrence, and lymph node metastasis [171]. Bi et al. implied that circ-BPTF is expressed at high levels in BC tissues, and its expression quantities are positively related with tumor size, recurrence, and lymph node metastasis [176]. In contrast, circ-ITCH is dramatically downregulated in BC tissues, and its quantities correlate negatively with tumor size, and stage [118]. Circ_0071196 has been revealed to be upregulated in BC tissues and cell lines, and high circ_0071196 expression quantities are positively related with tumor size, grade, and distant metastasis [177]. Similarly, circRGNEF is upregulated in BC tissues, and its quantities are correlated with tumor size, grade, and lymph node metastasis [104]. In addition, circZFR, circTFRC, hsa_circ_0068871, circ_0067934, circPTK2, circINTS4, circCEP128, circSEMA5A, circVANGL1, andcircEHBP1 [31,106,107,121,126,127,162,169,178,179], which are overexpressed in BC tissues, and ciRs-6, hsa_circ_0077837, circCDYL, circFUT8, circPTPRA, circ_0071662, circFOXO3, circFAM114A2, circUBXN7, and circRBPMS [111,116,130,133,141,144,170,180,181,182], which are downregulated in BC tissues, are also reported to be associated with many clinicopathological features in BC (Table 4).

## 6. Limitations and Prospects

Currently, there is growing interest in circRNAs and in gaining a better understanding of their roles in BC. However, the clinical use of circRNAs in BC remains largely unexplored and further studies are required before they can be integrated into clinical practice. Unlike prostate cancer, bladder cancer has no specific serum tumor markers like prostate specific antigen (PSA). Its diagnosis relies largely on clinical manifestations and imaging examination. This requires us to pay attention to find highly effective and sensitive biomarkers. In addition, the continuous development of modern medicine has enabled the transformation of oncology therapy from traditional treatment to a targeted treatment. Targeted treatment can specifically kill tumor cells without damaging normal peritumoral cells. However, resistance to targeted drugs is emerging in tumor cells. The mechanism of drug resistance relates to circRNAs, which provides a direction for further research.

In recent years, numerous studies have reported that after certain circRNAs were overexpressed or downregulated, the function of tumor cell lines were affected to varying degrees. Certain processes were reversed or enhanced partially, emphasizing the potential ability of circRNAs in the adjustment of tumor diseases, which is of great importance for clinical application.

There are still many obstacles to overcome in the clinical application of circular RNAs. First of all, we can identify a large number of differentially expressed circRNAs in cancer tissues and adjacent tissues using high-throughput sequencing, which is a credible approach that depends on the quality of RNA samples [183], but their functions and mechanisms have not been studied. Most usable RNA sequencing databases are pretreated via a poly (A) depuration step which can eliminate circRNAs, resulting in incomplete discovery of circRNAs. Furthermore, this technique is usually confined to a small sample size and limited tissue types, which is difficult for experienced pathologists to conduct an evaluation of [184]. Moreover, qRT-PCR has been used in most studies instead of Northern blotting to identify and verify circRNAs differentially expressed in RNA-seq due to its relative convenience and efficiency. However, certain circRNAs cannot be distinguished due to the occupation of different qRT-PCR primers on the back-splicing junction [185]. In qRT-PCR, circRNAs can only be differentiated from exon iteration by RNase or poly (A) enrichment pretreatment [186]. It has confirmed that several parental genes can encode or splice circRNAs, but the underlying associations and functional relationships between them are usually not as relevant as expected. According to the standardization of biomarkers, only a small part of the differentially expressed circRNAs meet the conditions, and only a small number of them have completed experiments in vivo and in vitro to prove their roles in tumor diagnosis, treatment and prognosis. In addition, there is no standard naming rules for circRNAs, which may lead to repeated studies on the same circRNA. Fortunately, many bioinformatics algorithms have been developed and progressively applied to study circRNAs. Certain algorithms call for gene annotation tables, while others need to be studied from scratch, which can lessen the existence of false positives and prevent the omission of unannotated specific circRNAs due to a lack of uniform naming [187]. Through the prediction from bioinformatics algorithms, the hypothetical interaction between circRNAs and miRNAs can be checked by stoichiometric relationship, but this method is difficult to conduct because a great number of cells that differentially express circRNAs and miRNAs need to be analyzed simultaneously. Meanwhile, luciferase reporter analysis, as a representative of classical methods to determine direct binding, is seldom used in the studies of circRNAs. The location and metabolism of circRNAs have also been rarely reported. A large number of hypotheses have been proposed, including the secretion of circRNAs into exosomes, which need more studies to verify. Many investigations have concentrated on the function of circRNAs as miRNA sponges while ignoring other underlying features of circRNAs. Recent research has challenged the conventional wisdom, for instance, the absence of MREs do not affect the inhibitory actions of circRNAs on specific target miRNAs and there are raising questions about the structural foundation of the miRNA “sponge” [188]. No translatable circRNAs were identified in osteosarcoma by ribosome footprint detection, which questions the reliability of the protein coding capacity [189]. Since circRNA research is still in its infancy, these multifarious problems provided by these challenges deserve deeper explorations. With innovations of molecular biological techniques, people believe that those challenges will eventually be overcome, possibly leading to breakthroughs in circRNA research.

The epigenetic changes in human malignancies are primarily histone modifications, gene promoter hypermethylation, global genomic hypomethylation and modified miRNA expression patterns [190]. Thus, the differential expressions of circRNAs in BC associated with epigenetic changes in the genome will be an interesting direction to investigate. Furthermore, it has been shown that in colorectal cancer, the expression of circRNAs is modulated by mutant cis-acting elements. Whether the same modalities of regulation exist in BC and their roles in circRNAs biogenesis deserve further exploration. The post-transcriptional chemical modification of RNAs may be essential for their functionality and stability. Therefore, the exploration of chemical modification of circRNAs to achieve the adjustment of their functions is also a possible research direction. With the establishment of databases such as CircBase, CircNet and DeepBasev2.0, we can improve the naming system to unify the IDs of circRNAs [60,62,191]. Databases need to be updated in an immediate manner to effectively evade errors caused by bioinformatics algorithm and improve the validity of research hypotheses. In addition, there are more circRNAs detection methods developed. On top of qRT-PCR and Northern blotting, in situ hybridization (ISH) and fluorescence in situ hybridization (FISH) address the expression and distribution of circRNAs by providing direct visualization of spatial information [37]. The separation and sequencing of single cell are more suitable for detecting differentially expressed circRNAs in a cohort of cells or tissues. In terms of probing the associations between circRNAs and parental genes, clustered regularly interspaced short palindromic repeats/Cas9 (CRISPR-cas9) is more advantageous [192]. As research techniques and horizons continue to be updated and expanded, our understanding of circRNAs in BC will be further enhanced.

## 7. Conclusions

With the significant progress of RNA research, circRNAs have attracted extensive attention of researchers and have gradually become an emerging frontier in cancer research. A soaring number of circRNA transcripts have been discovered, and certain circRNAs have been proven to be functional ncRNAs associated with malignant phenotypes and clinical manifestations. In this review, we not only introduced the biogenesis and function of circRNAs, but also the role of circRNAs as clinical biomarkers in the diagnosis, prognosis, and drug resistance of BC, and further discuss the functions and significance of circRNAs in BC. Notably, current research primarily reveals the role of circRNAs as miRNA sponges, while other potential functions of circRNAs in BC need to be further studied. Although research in circRNA regulation of bladder cancer is still in its infancy and many questions remain unanswered, we believe that circRNAs can provide a new avenue for tumor treatment.

## Figures and Tables

**Figure 1 cancers-13-04618-f001:**
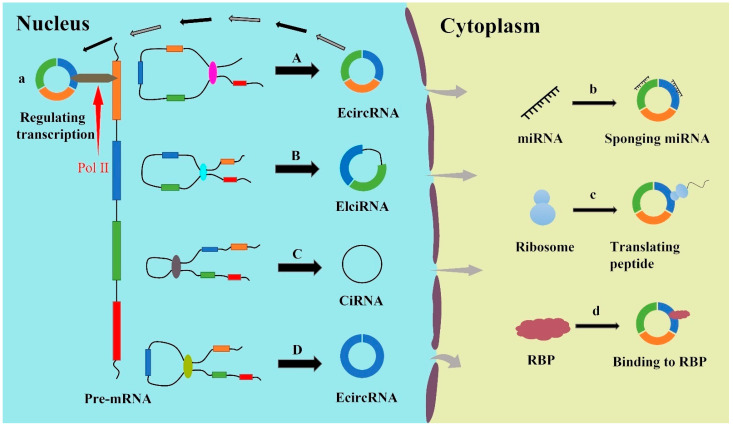
Biogenesis and function of circRNAs .A. EcircRNA is generated from linking the downstream 5′ terminal of an exon to the upstream 3′ terminal of another exon and cyclizing. B. EIciRNA is formed by intron pairing-driven circularization, which is induced by reverse complementary sequences. C. The formation of CiRNA depends on conserved sequences near the spliceosome. D. EcircRNA can also be generated from a single exon, where the 5′ terminal of one exon is connected to the 3′ terminal of the same exon. a. Regulation of gene transcription. b. MiRNA sponges. c. Translation into peptides and/or proteins. d. Interaction with functional proteins.

**Figure 2 cancers-13-04618-f002:**
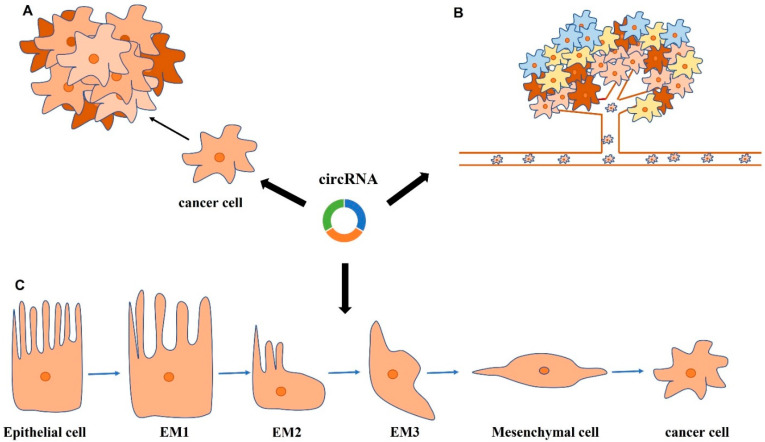
CircRNAs regulate cell proliferation, migration and invasion, and modulate epithelial-mesenchymal transition (EMT). (**A**). Roles of circRNAs in cell proliferation. (**B**). Roles of circRNAs in cell migration and invasion. (**C**). Roles of circRNAs in modulating EMT.

**Table 1 cancers-13-04618-t001:** CircRNAs and their mechanisms in bladder cancer progression.

CircRNA	Circbase ID	Gene Symbol	Expression	Function	Molecular Mechanism	Target Gene/Pathway	Reference
CircUVRAG	Hsa_circ_0023642	UVRAG	Upregulated	Promoting cell proliferation and migration	Sponging for miR-223	FGFR2	[67]
CircRGNEF	Hsa_circ_0072995	RGNEF	Upregulated	Promoting cell proliferation and invasion	Sponging for miR-548	KIF2C	[104]
CircRIP2	Has_circ_0005777	RIP2	Upregulated	Promoting cell proliferation and metastasis	Sponging for miR-1305	Tgf-β2/smad3	[105]
Circ_0068871	Hsa_circ_0068871	-	Upregulated	Promoting cell proliferation and migration	Sponging for miR-181a-5p	FGFR/STAT3	[106]
CircCEP128	Hsa_circ_0102722	-	Upregulated	Promoting cell proliferation and inhibiting cell apoptosis	Sponging for miR-145-5p	SOX11	[107]
CircDOCK1	Hsa_circ_0020394	-	Upregulated	Promoting cell proliferation and migration	Sponging for miR-132-3p	SOX5	[108]
Circ_0068307	Hsa_circ_0068307	-	Upregulated	Promoting cell proliferation and migration	Sponging for miR-147	c-Myc	[109]
Circ_0008532	Hsa_circ_0008532	-	Upregulated	Promoting cell migration, invasion, and angiogenesis	Sponging for miR-155-5p/miR-330-5p	MTGR1	[110]
CircFOXO3	Hsa_circ_0006404	FOXO3	Downregulated	Inhibiting cell proliferation, migration and invasion	Sponging for miR-9-5p	TGFBR2	[111]
CircFOXO3	Hsa_circ_0006404	FOXO3	Downregulated	Promoting promoted cell apoptosis	Sponging for miR-191-5p	-	[112]
CircBCRC-3	Hsa_circ_0001110	BCRC-3	Downregulated	Inhibiting cell proliferation, and promoting cell cycle arrest	Sponging for miR-182-5p	P27	[113]
CircCdr1as	Hsa_circ_0001946	CDR	Downregulated	Inhibiting cell proliferation, migration and invasion	Sponging for miR-135a	P21	[114]
CircNR3C1	Hsa_circ_0001543	NR3C1	Downregulated	Inhibiting cell proliferation and cell cycle progression	Sponging for miR-27a-3p	Cyclin D1	[115]
CircPTPRA	Hsa_circ_0006117	PTPRA	Downregulated	Inhibiting cell proliferation	Sponging for miR-636	KLF9	[116]
CircSLC8A1	Hsa_circ_0000994	SLC8A1	Downregulated	Inhibiting cell proliferation, migration and invasion	Sponging for miR-130b/miR-494	PTEN	[117]
CircITCH	Hsa_circ_0001141	ITCH	Downregulated	Inhibiting cells proliferation, migration, invasion and metastasis	Sponging for miR-17/miR-224	P21/PTEN	[118]
CircBCRC4	Hsa_circ_0001577	RANBP9	Downregulated	Inhibiting cell viability and promoting cell apoptosis	Sponging for miR-101	EZH2	[119]
CircVANGL1	Hsa_circ_0002623	VANGL1	Upregulated	Promoting cell proliferation, migration and invasion	Sponging for miR-1184	IGFBP2	[120]
CircVANGL1	Hsa_circ_0002623	VANGL1	Upregulated	Promoting cell proliferation, migration, and invasion	Sponging for miR-605-3p	VANGL1	[121]
CircCEP128	Hsa_circ_0102722	-	Upregulated	Promoting cell proliferation and migration, inhibiting cell apoptosis and cell cycle arrest	Sponging for miR-145-5p	MAPK/MYD88	[122]
Circ_0058063	Hsa_circ_0058063	-	Upregulated	Promoting cell proliferation and migration, inhibiting cell apoptosis	Sponging for miR-145-5p	CDK6	[123]
Circ_0058063	Hsa_circ_0058063	-	Upregulated	Promoting cell proliferation and invasion, inhibiting apoptosis	Sponging for miR-486-3p	FOXP4	[124]
CircTCF25	Hsa_circ_0041103	-	Upregulated	Promoting cell proliferation and migration	Sponging for miR-107/miR-103-3p	CDK6	[125]
CircTFRC	Has_circ_0001445	TFRC	Upregulated	Promoting cell proliferation and invasion	Sponging for miR-107	TFRC	[126]
CircINTS4	Hsa_circ_0002476	INTS4	Upregulated	Promoting cell proliferation, migration, cell cycle and apoptosis	Sponging for miR-146b	CARMA3/NFκB/P38 MAPK	[127]
CircKIF4A	Hsa_circ_0007255	-	Upregulated	Promoting cell proliferation and colony-formation ability	Sponging for miR-375 and miR-1231	NOTCH/PI3K/AKT	[128]
Circ_0071662	Hsa_circ_0071662	TPPP1	Downregulated	Inhibiting cell proliferation and invasion	Sponging for miR-146b-3p	HPGD/NF2	[129]
CircFAM114A2	Hsa_circ_0001546	FAM114A2	Downregulated	Inhibiting cell proliferation, migration and invasion	Sponging for miR-762	ΔNP63	[130]
Circ_0091017	Hsa_circ_0091017	-	Downregulated	Inhibiting cell proliferation, migration and invasion	Sponging for miR-589-5p	-	[131]
Circ_0002024	Hsa_circ_0002024	-	Downregulated	Inhibiting cell proliferation, migration and invasion	Sponging for miR-197-3p	-	[132]
CircUBXN7	Hsa_circ_0001380	UBXN7	Downregulated	Inhibiting cell proliferation, migration and invasion	Sponging for miR-1247-3p	B4GALT3	[133]
CircFNDC3B	Hsa_circ_0006156	FNDC3B	Downregulated	Inhibiting cell proliferation, migration and invasion	Sponging for miR-1178-3p	G3BP2/SRC/FAK	[134]
Circ_0023642	Hsa_circ_0023642	UVRAG	Downregulated	Inhibiting cell invasion	Sponging for miR-490-5p	EGFR	[135]
CircPTPRA	Hsa_circ_0006117	PTPRA	Downregulated	Inhibiting cell invasion, metastasis and cell cycle	Interacting with IGF2BP1	M6A-modified RNAs	[136]
Circ_0006332	Hsa_circ_0006332	MYBL2	Upregulated	Promoting cell proliferation, colony formation and invasion	Sponging for miR-143	MYBL2/EMT	[137]
CircRIMS1	Hsa_circ_0132246	-	Upregulated	Promoting cell proliferation, migration and invasion	Sponging for miR-433-3p	CCAR1/EMT	[32]
CircPRMT5	Hsa_circ_0031250	PRMT5	Upregulated	Promoting cell migration and invasion	Sponging for miR-30c	EMT	[138]
CircMYLK	Hsa_circ_0002768	MYLK	Upregulated	Promoting cell proliferation, migration and angiogenesis	Sponging for miR-29a	VEGFA/VEGFR2 and Ras/ERK, and EMT	[139]
Circ_100984	Hsa_circ_100984	-	Upregulated	Promoting cell proliferation, migration and invasion	Sponging for miR-432-3p	c-Jun/YBX-1/β-catenin and EMT	[140]
CircRBPMS	Hsa_circ_0006539	RBPMS	Downregulated	Inhibiting cell proliferation and metastasis	Sponging for miR-330-3p	RAI2/ERK/EMT	[141]
CircST6GALNAC6	Hsa_circ_0088708	ST6GALNAC6	Downregulated	Inhibiting cell proliferation, migration, invasion	Sponging for miR-200a-3p	STMN1/EMT	[142]
Circ_0000629	Hsa_circ_0000629	-	Downregulated	Inhibiting cell migration, invasion and growth	Sponging for miR-1290	CDC73/EMT	[143]
CircPICALM	Hsa_circ_0023919	PICALM	Downregulated		Sponging for miR-1265	STEAP4/pFAK-Y397/EMT	[144]

**Table 2 cancers-13-04618-t002:** circRNAs and their values in bladder cancer chemo-sensitization and drug-resistance.

CircRNA	CircBase ID	Gene Symbol	Expression	Clinical Value	Molecular Mechanism	Target Gene/Pathway	Reference
CircCdr1as	Hsa_circ_0001946	CDR1	Downregulated	Promoting cisplatin sensitivity	Sponging for miR-1270	APAF1	[155]
CircHIPK3	Hsa_circ_0000284	HIPK3	Downregulated	Promoting gemcitabine sensitivity	Sponging for miR-558	HPSE, VEGF, MMP9	[156]
CircELP3	Hsa_circ_0001785	ELP3	Upregulated	Promoting cisplatin resistance	Hypoxia-elevated	cancer stem-like cells	[157]
CircFNTA	Hsa_circ_0084171	FNTA	Upregulated	Promoting cisplatin resistance	Sponging for miR-370-3p	KRAS	[158]
Circ_102336	Hsa_circ_102336	TAF4B	Upregulated	Promoting cisplatin resistance	Sponging for miR-515-5p	ATP-binding cassette (ABC) transporters and apoptosis pathways	[159]
Circ_0000285	Hsa_circ_0000285	HIPK3	Downregulated	Promoting cisplatin sensitivity	Unknown	Unknown	[160]
Circ_103809	Hsa_circ_0072088	ZFR	Upregulated	Promoting chemo-resistance	Sponging for miR-516a-5p	FBXL18	[161]

**Table 3 cancers-13-04618-t003:** CircRNAs and their values in bladder cancer diagnosis and prognosis.

CircRNA	CircBase ID	Gene Symbol	Expression	Clinical Value	Molecular Mechanism	Target Gene/Pathway	Reference
CircVANGL1	Hsa_circ_0002623	VANGL1	Upregulated	Prognostic utility	Sponging for miR-605-3p	VANGL1	[121]
Circ_0067934	Hsa_circ_0067934	-	Upregulated	Prognostic utility	Sponging for miR-1304	Myc	[162]
CircASXL1	Hsa_circ_0001136	ASXL1	Upregulated	Prognostic and diagnostic utility	Unknown	unknown	[163]
CircGprc5a	Hsa_circ_02838	-	Upregulated	Prognostic utility	Unknown	Gprc5a protein	[164]
Circ_0001361	Hsa_circ_0001361	FNDC3B	Upregulated	Prognostic utility	Sponging for miR-491-5p	MMP9	[165]
Circ_0000285	Hsa_circ_0000285	HIPK3	Downregulated	Prognostic utility	Unknown	unknown	[160]
CircMTO1	Hsa_circ_0007874	MTO1	Downregulated	Prognostic utility	Sponging for miR-221	unknown	[166]
CircLPAR1	Hsa_circ_0087960	LPAR1	Downregulated	Prognostic utility	Sponging for miR-762	unknown	[167]
CircACVR2A	Hsa_circ_0001073	ACVR2A	Downregulated	Prognostic utility	Sponging for miR-626	EYA4	[168]
CircZFR	Hsa_circ_0072088	ZFR	Upregulated	Prognostic and diagnostic utility	Sponging for miR-377	ZEB2	[169]
CircFUT8	Hsa_circ_0003028	FUT8	Downregulated	Prognostic utility	Sponging for miR-570-3p	KLF10	[170]
CircZKSCAN1	Hsa_circ_0001727	ZKSCAN1	Downregulated	Prognostic utility	Sponging for miR-1178-3p	P21	[171]
CircPICALM	Hsa_circ_0023919	PICALM	Downregulated	Prognostic utility	Sponging for miR-1265	STEAP4	[144]
Circ_0018069	Hsa_circ_0018069	-	Downregulated	Prognostic and diagnostic utility	Sponging for miR23c, miR-34a-5p, miR-181b-5p, miR-454-3p and miR-3666	ErbB, Ras, Foxo, and the focal adhesion	[172]

**Table 4 cancers-13-04618-t004:** Relationships between circRNAs quantities and clinicopathologic features in BC.

CircRNA	CircBase ID	Gene Symbol	Expression	Tumor Stage	Tumor Grade	Tumor Size	Tumor Recurrence	Tumor Number	Reference
CircRIP2	Has_circ_0005777	RIP2	Upregulated	Yes	-	Yes	-	Yes	[105]
Circ_100146	Hsa_circ_100146	-	Upregulated	Yes	Yes	Yes	-	Yes	[173]
Circ_0006332	Hsa_circ_0006332	MYBL2	Upregulated	Yes	Yes	Yes	-	-	[137]
CircHIPK3	Hsa_circ_0000284	HIPK3	Downregulated	Yes	-	Yes	-	-	[174]
CircFNDC3B	Hsa_circ_0001361	FNDC3B	Downregulated	Yes	Yes	-	Yes	-	[165]
CircPICALM	Hsa_circ_0023919	PICALM	Downregulated	Yes	Yes	Yes	-	Yes	[144]
Circ_0137439	Hsa_circ_0137439	MDTH	Upregulated	Yes	Yes	-	-	-	[175]
CircSLC8A1	Hsa_circ_0000994	SLC8A1	Downregulated	Yes	Yes	Yes	-	-	[117]
Circ_0058063	Hsa_circ_0058063	-	Upregulated	Yes	Yes	-	-	Yes	[124]
CircZKSCAN1	Hsa_circ_0001727	ZKSCAN1	Downregulated	Yes	Yes	-	-	Yes	[171]
CircBPTF	Hsa_circ_0000799	BPTF	Downregulated	Yes	-	Yes	Yes	-	[176]
CircITCH	Hsa_circ_0001141	ITCH	Downregulated	Yes	-	Yes	-	Yes	[118]
Circ0071196	Hsa_circ_0071196	-	Upregulated	Yes	Yes	Yes	-	-	[177]
CircRGNEF	Hsa_circ_0072995	RGNEF	Upregulated	Yes	Yes	Yes	-	-	[104]
CircZFR	Hsa_circ_0072088	ZFR	Upregulated	Yes	Yes	Yes	Yes	-	[169]
CircTFRC	Has_circ_0001445	TFRC	Upregulated	Yes	Yes	Yes	-	-	[126]
Circ_0068871	Hsa_circ_0068871	-	Upregulated	Yes	-	Yes	-	-	[106]
Circ_0067934	Hsa_circ_0067934	-	Upregulated	Yes	-	Yes	-	-	[39]
CircPTK2	Hsa_circ_0003221	PTK2	Upregulated	Yes	Yes	Yes	-	-	[178]
CircINTS4	Hsa_circ_0002476	INTS4	Upregulated	Yes	Yes	Yes	-	-	[127]
CircCEP128	Hsa_circ_0102722	-	Upregulated	Yes	-	Yes	-	-	[107]
CircSEMA5A	Hsa_circ_0071820	SEMA5A	Upregulated	Yes	Yes	Yes	-	Yes	[179]
CircVANGL1	Hsa_circ_0002623	VANGL1	Upregulated	Yes	Yes	Yes	-	-	[121]
CircEHBP1	Hsa_circ_0005552	-	Upregulated	Yes	Yes	Yes	-	-	[31]
CiRs_6	Hsa_circ_0006260	SLC41A2	Downregulated	Yes	Yes	Yes	-	Yes	[180]
Circ_0077837	Hsa_circ_0077837	-	Downregulated	Yes	Yes	Yes	-	Yes	[181]
CircCDYL	Hsa_circ_0008285	-	Downregulated	Yes	Yes	Yes	-	-	[182]
CircFUT8	Hsa_circ_0003028	FUT8	Downregulated	Yes	Yes	Yes	-	Yes	[170]
CircPTPRA	Hsa_circ_0006117	PTPRA	Downregulated	Yes	Yes	Yes	-	Yes	[116]
Circ_0071662	Hsa_circ_0071662	TPPP1	Circ_0071662	Yes	Yes	Yes	-	-	[129]
CircFOXO3	Hsa_circ_0006404	FOXO3	Downregulated	Yes	Yes	Yes	-	-	[111]
CircFAM114A2	Hsa_circ_0001546	FAM114A2	Downregulated	Yes	Yes	Yes	-	-	[130]
CircUBXN7	Hsa_circ_0001380	UBXN7	Downregulated	Yes	Yes	Yes	-	-	[133]
CircRBPMS	Hsa_circ_0006539	RBPMS	Downregulated	Yes	Yes	Yes	-	-	[87]

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
