# Peer review of "The Emerging Functions of Circular RNAs in Bladder Cancer"

_cancers, 2021, doi:10.3390/cancers13184618_

Round 1
Reviewer 1 Report
The manuscript is presenting all relevant data and knowledge on the topic of circular RNAs in bladder cancer.
It is well organized and it is covering all the necessary aspects of the topic.
The most interesting part of the review is part 6 – limitations and prospects. All other parts are written in a more informative nature with stating pure facts from the publications which are also presented in tables.
Unfortunately, there is not much novelty in the review as there is a very similar review from Cheng, F et al published recently (July 2021) in Frontiers in Cell and Developmental Biology (https://www.frontiersin.org/articles/10.3389/fcell.2021.666863/full).
Authors should consider rewriting the manuscript and focusing in more detail on specific aspects of the topic.
Minor revision
- Few references are incorrect - ref.103 is incorrect in table 1., ref. 136 and 137 are incorrect in table 3., ref. 147 is incorrect in table 4
- Figure 1. It is hard to follow the figure. Letters should be more “visible” (e.g. bigger letters or in different colours) and maybe in more intuitive order (e.g. nucleus at top of the figure, and letter from the top left to the bottom right).
Author Response
Reviewer 1
The manuscript is presenting all relevant data and knowledge on the topic of circular RNAs in bladder cancer.
It is well organized and it is covering all the necessary aspects of the topic.
The most interesting part of the review is part 6 – limitations and prospects. All other parts are written in a more informative nature with stating pure facts from the publications which are also presented in tables.
Response: Thank you for your time to review my manuscript and your positive comments on the review. Indeed, we think it is important to have inputs and opinions in a review, and readers would be interested in knowing the opinions of the authors.
Unfortunately, there is not much novelty in the review as there is a very similar review from Cheng, F et al published recently (July 2021) in Frontiers in Cell and Developmental Biology (https://www.frontiersin.org/articles/10.3389/fcell.2021.666863/full). Authors should consider rewriting the manuscript and focusing in more detail on specific aspects of the topic.
Response: Thank you for pointing out this newly published review by Cheng and colleagues. Cheng and colleagues’ review and ours focused on different areas of knowledge related to circRNAs in bladder cancer. For example, in Cheng and colleagues’ review, they have 4 sections (subtitles) to talk about “FUNCTIONS OF CIRCULAR RNAs”. In our review, we only have one paragraph to briefly introduce “Functions of circRNAs” (no subtitle). This has allowed us to have more rooms to describe in greater detail for the “3. Expression and biological functions of circRNAs in BC”.
In addition, we have a large section to describe the “5. Relationships between circRNAs quantities and clinicopathologic features in BC”. We spent much time to summarize the relationship between quantity and pathology.
In section 6. Limitations and prospects and section 7. Conclusion, these are the inputs and opinions of the authors, both reviews are different.
Thus, while we acknowledge this newly published review (now cited in the reference list), we followed you suggestions to focus on different areas to decrease the similarity to this review.
Minor revision
Few references are incorrect - ref.103 is incorrect in table 1., ref. 136 and 137 are incorrect in table 3., ref. 147 is incorrect in table 4
Response: Thank you for your careful review of my manuscript. We have carefully reviewed the references and corrected them for errors. The modified tables have been updated and re-uploaded.
Figure 1. It is hard to follow the figure. Letters should be more “visible” (e.g. bigger letters or in different colours) and maybe in more intuitive order (e.g. nucleus at top of the figure, and letter from the top left to the bottom right).
Response: Thank you for your suggestion. Combined with the suggestion provided by other reviewers, we have generated new figures and re-uploaded them.
Reviewer 2 Report
The authors describe the new field of Circular RNAs (circRNAs) in the context of bladder cancer. The description of circRNAs is sufficient, however it would be good to describe the lariat formation a little bit more in detail. This would enable a better understanding for readers without further knowledge of circRNAs.
As a second small revision point, I would suggest not to decorate 2-dimensional illustrations with 3-dimensional figures. this is not particularly aesthetic and is only unpleasantly conspicuous in the illustration.
However, this is a nice and well realized review about the possible multiple functions of circRNAs in cancers.
Author Response
Reviewer 2
The authors describe the new field of Circular RNAs (circRNAs) in the context of bladder cancer.
The description of circRNAs is sufficient, however it would be good to describe the lariat formation a little bit more in detail. This would enable a better understanding for readers without further knowledge of circRNAs.
Response: Thank you for your suggestion. Lariat formation is an interesting and important process in the formation of circRNA. We have modified the manuscript to have this information understood clearly. We have changed the text as follows:
“Lariat-driven circularization is a kind of exon skipping process. Pre-mRNA splicing is commonly known to be a two-step ester exchange reaction38. The adenosine hydroxyl group located at the 2’ branch point within the intron that is to be spliced attacks the intron upstream of the 5’ end in a nucleophilic manner. This produces a lariat intermediate closed covalently through the 2’ to 5’ phosphodiester bond. Then, the 3’ hydroxyl group of the upstream exon is free to attack the 5’ phosphate of the downstream exon, splicing out the intron completely and allowing the exons to be attached as a linear coding sequence38. After being processed through lariat model, pre-mRNAs form a lariat intermediate containing exons with the help of circular introns during internal splicing39, 40. In the lariat intermediate, the 5ʹ terminal donor of an exon combines with the 3ʹ terminal acceptor of another exon, and then forms an EcircRNA by eliminating introns between exons.” (Please see Page 3, blue label)
As a second small revision point, I would suggest not to decorate 2-dimensional illustrations with 3-dimensional figures. this is not particularly aesthetic and is only unpleasantly conspicuous in the illustration.
However, this is a nice and well realized review about the possible multiple functions of circRNAs in cancers.
Response: Thank you for your suggestion. Combined with the suggestions from the other reviewers, we have generated new figures and re-uploaded them to the revised manuscript.
Reviewer 3 Report
This is a particularly useful and well-written review article. It clearly summarizes the recent findings on circRNAs in bladder cancer, from a functional perspective. I strongly recommend the acceptance of this article, provided that the following comments are properly addressed:
- In the section “2.3 Functions of circRNAs”, the authors state: “Dysregulation of miRNA-mediated mRNA and correlative signaling pathways are closely interrelated to cancer progression and therapy resistance. CircRNAs function as miRNA sponges or competing endogenous RNA (ceRNAs) to repress miRNA.” However, they need to clearly also present the conclusion that circular RNAs are, thus, critical regulators of the major signaling pathways involved in cancer progression [Papatsirou et al., Cancers (Basel). 2021 Jun 1;13(11):2744. doi: 10.3390/cancers13112744].
- In the section “2.3 Functions of circRNAs”, the authors state: “However, chemotherapy resistance often leads to tumor recurrence and progression. Several studies have reported the relationship between circRNAs and chemotherapeutic resistance, suggesting that circRNAs may be potential therapeutic targets”; the authors are advised to add a reference here. An absolutely relevant review to cite here is: “The role of circular RNAs in therapy resistance of patients with solid tumors” by Papatsirou et al. [Per Med. 2020 Nov;17(6):469-490. doi: 10.2217/pme-2020-0103].
Author Response
Reviewer 3
This is a particularly useful and well-written review article. It clearly summarizes the recent findings on circRNAs in bladder cancer, from a functional perspective. I strongly recommend the acceptance of this article, provided that the following comments are properly addressed:
Response: Thank you for your very careful review of my manuscript. For the errors listed below, we have now made the corrections and checked the manuscript carefully to avoid such errors. We have also followed your suggestions and include the references as follows.
In the section “2.3 Functions of circRNAs”, the authors state: “Dysregulation of miRNA-mediated mRNA and correlative signaling pathways are closely interrelated to cancer progression and therapy resistance. CircRNAs function as miRNA sponges or competing endogenous RNA (ceRNAs) to repress miRNA.” However, they need to clearly also present the conclusion that circular RNAs are, thus, critical regulators of the major signaling pathways involved in cancer progression [Papatsirou et al., Cancers (Basel). 2021 Jun 1;13(11):2744. doi: 10.3390/cancers13112744].
Response: Thank you for your suggestions. We totally agree that circRNAs are key regulators of major signaling pathways involved in cancer progression and have revised the manuscript as follows.
“CircRNAs have been identified as critical regulators of the major signaling pathways involved in cancer progression64. Dysregulation of miRNA-mediated mRNA and correlative signaling pathways are closely interrelated to cancer progression and therapeutic resistance. Numerous studies have reported that circRNAs can function as miRNA sponges or competing endogenous RNA (ceRNAs) to repress miRNA activities65-68.” (Please see Page 5, blue label)
In the section “2.3 Functions of circRNAs”, the authors state: “However, chemotherapy resistance often leads to tumor recurrence and progression. Several studies have reported the relationship between circRNAs and chemotherapeutic resistance, suggesting that circRNAs may be potential therapeutic targets”; the authors are advised to add a reference here. An absolutely relevant review to cite here is: “The role of circular RNAs in therapy resistance of patients with solid tumors” by Papatsirou et al. [Per Med. 2020 Nov;17(6):469-490. doi: 10.2217/pme-2020-0103
Response: Thank you for your suggestions. We have cited the reference mentioned in the revised manuscript.
Round 2
Reviewer 1 Report
The manuscript is updated and improved according to all reviewers comments and now is ready for publishing.